# “*I Don’t Believe in Age; I Believe in Staying Enthusiastic*”: An Exploratory Qualitative Study into Recruitment Strategies Stimulating Middle-Aged and Older Adults to Join Physical Activity Interventions

**DOI:** 10.3390/geriatrics9030080

**Published:** 2024-06-13

**Authors:** Janet M. Boekhout, Rieteke Hut, Lilian Lechner, Denise A. Peels

**Affiliations:** Department of Health Psychology, Faculty of Psychology, Open University of The Netherlands, 6401 DL Heerlen, The Netherlands; rieteke.hut@ou.nl (R.H.); lilian.lechner@ou.nl (L.L.); denise.peels@ou.nl (D.A.P.)

**Keywords:** physical activity, older adults, middle aged, interventions, recruitment strategies, implementation, healthy aging

## Abstract

Many middle-aged and older adults (MAOAs) do not engage in sufficient physical activity (PA), despite its well-documented benefits for healthy aging. Existing PA interventions often fail to reach or engage the target population effectively. This study investigates MAOAs’ preferences for recruitment strategies to optimize the reach and uptake of PA interventions, thereby enhancing their impact on healthy aging and public health. Qualitative interviews were conducted with 39 MAOA participants (69% female, mean age = 69.46, SD = 7.07), guided by McGuire’s Theory on Persuasive Communication. Factors related to the source, message content, channel, receiver characteristics and target behavior of recruitment strategies were analyzed. Our findings suggest a preference for trustworthy sources (e.g., healthcare professionals over commercial entities) and positive, non-ageist messaging. MAOAs vary in their channel preferences but emphasize the importance of personalization. Despite heterogeneity, MAOAs commonly perceive themselves as sufficiently active, indicating a need for improved knowledge on what constitutes sufficient PA, as well as easy enrollment or trying out interventions. Tailoring recruitment strategies to diverse MAOA segments based on age seems crucial for effective engagement. Future research could explore quantitative research into how communication factors relate to various target population characteristics.

## 1. Introduction

Physical activity (PA) in middle-aged and older adults (MAOAs) is a topic of major interest and concern for governments worldwide, given its high benefits for public health [1,2,3]. In recent decades, a wide range of interventions that aim to stimulate PA in MAOAs have been developed (including eHealth interventions), and ample evidence has demonstrated the effectiveness of such interventions [4,5,6,7,8,9]. The impact of PA interventions on public health, however, is not only determined by their effectiveness, but also on their reach and use; if MAOAs are not made aware of the existence of an intervention, or if they are not prompted to participate in an intervention, the intervention will not have the impact on PA, as previously demonstrated in its efficacy trial [10,11]. Although the issue of high non-participation in PA interventions among MAOAs is well documented, most research so far seems to focus on demographic characteristics as possible explanations for non-participation, but not on explanations that may lie within how communication about the intervention is shaped [12,13,14,15,16,17]. More knowledge in this area is essential because unless PA interventions are adopted on a larger scale by MAOAs than they have been so far, these interventions will not sufficiently contribute to positive outcomes in the process of healthy aging. In an aging society, the large-scale implementation and adoption of healthy aging interventions, such as PA interventions, are essential in order to keep public health affordable.

Recruitment strategies can be defined as types of persuasive communication designed to inform specific target populations about the existence of a particular intervention and to encourage the target population to participate in that intervention. Recruitment strategies aim to attract and motivate potential participants to engage in a particular behavior or intervention. Examples include selecting a source to deliver messages about interventions that potential participants find trustworthy or emphasizing in messages the benefits of PA that are important to MAOAs. For this, attitudes and actions of the target population regarding PA and PA interventions need to be addressed. A guiding framework in persuasion communication is provided by McGuire’s framework for persuasive communication [18]. Most research so far, however, seems to have been conducted by applying McGuire’s framework in designing public health campaigns [19,20,21]. To our knowledge, research into the utility of this framework when designing communications around PA interventions targeting MAOAs is scarce. We postulate that, as both persuasive communication in public health campaigns and recruitment strategies strive to influence attitudes and actions of large groups of people, McGuire’s framework is also suitable to be applied in designing recruitment strategies. The aim of the current paper is to use this framework to explore MAOAs’ perceptions of recruitment strategies for PA interventions in general. The latter is especially relevant in aging societies, as such interventions have the potential to reach a large part of the target population at low costs [22,23], but adoption may still be challenging for MAOAs, especially for the oldest old [24,25].

McGuire’s framework [18] enables a thorough evaluation of potential communications and can thus facilitate designing recruitment strategies that will lead to the desired behavior. The framework consists of a matrix with communication factors and response steps. The communication factors can be seen as components out of which the recruitment strategies can be constructed in order to change attitudes and actions. The response steps are consecutive stages of information processing, including attentional, cognitive and decision-making steps, that the recruitment strategies must elicit in the target population for the persuasive impact to occur. As most of the communication factors are within the influence of the intervention owner or intervention implementer, these can be attuned. When attuned correctly, the target population will go through the stages of the response steps with the result of performing the desired behavior. The communication factors are (1) the source of the communication, i.e., the characteristics of the communicator that conveys the message, which comprise the number of sources, unanimity, attractiveness, credibility and demographics; (2) the message factors in the communication, i.e., the type of appeal and information, repetitiveness, inclusion and omission; (3) the channel, i.e., the media used to distribute the communication, such as modality, directness and context; (4) the receiver of the communication, i.e., the attributes and traits of the individuals who are receiving the persuasive message, such as demographics, prior experience, attitudes, motivation, emotion, personality and lifestyle; and (5) the target behavior, i.e., the type of actions the target population will perform, such as immediacy or delay, prevention or cessation, and direct action or immunization [18]. Combined, the above results in information on who (source) says what (message) how (channel) to whom (receiver) regarding which action (i.e., the target behavior of actually participating in an intervention).

As a scoping review [26] has recently shown, the matter of engaging target populations in PA interventions needs further research. The goal of this qualitative study is to use McGuires’ Persuasion Communication Framework to enhance our understanding of the perceptions of MAOAs regarding recruitment strategies. Our research question is how persuasion communication knowledge can deepen our understanding of how communication on PA interventions for MAOAs should optimally be shaped. Our objectives are to identify the features of the source, message factors, channels, receiver characteristics and target behavior that MAOAs prefer in recruitment strategies. Our study can provide insight into how MAOAs can best be reached for interventions and can thus contribute to the improved implementation of PA interventions. When PA interventions are implemented optimally, they can have a better impact on the process of healthy aging.

## 2. Materials and Methods

### 2.1. Design and Procedure

This study used a qualitative design for gaining in-depth knowledge on the perceptions of MAOAs regarding communication about PA interventions. Semi-structured individual interviews were conducted between August and November 2022. Participants were provided with an information letter providing detailed information about the study. The information letter described that participation was voluntary and confidential and that it could be stopped at any time. All participants gave informed consent for inclusion prior to the start of the interview. We adhered to the Consolidated Criteria for Reporting Qualitative Research (COREQ) in reporting this study [27].

The research team consisted of three principal researchers (R.H., J.B. and D.P.) supported by three master student researchers (K.A., M.H. and R.G.). All of the researchers are trained at a master’s level in qualitative research. The three main researchers have previous experience in conducting qualitative research. Their previous research is mainly in the field of PA in older adults, the well-being of older adults and the implementation of PA interventions. 

Participants met the following inclusion criteria: 50 years or older with a good understanding of the Dutch language. No other exclusion criteria were applied. Recruitment was conducted by way of a convenience sample: potential participants who met the inclusion criteria were selected in the personal and professional networks of the researchers and approached by phone, by email or in person. By using the snowball method, interviewed participants were asked if they could refer to other potential participants. Purposive sampling was used after 39 participants were included. In order to obtain the perspectives of a sufficiently varying sample regarding age and sex, a special effort was made in the recruitment to obtain a sufficiently even distribution of participants aged under and over 65, as this is the age in The Netherlands when the majority of people retire [28] and PA patterns tend to change [29].

A total of 68 participants were invited to participate, of which 39 were interviewed (25 in individual interviews and 7 in a double interview with married couples). The other 29 either did not respond to the invitation (*n* = 17), responded not to be interested or to lack time (*n* = 7), did not want to be voice recorded (*n* = 1), or dropped out of the research before the scheduled interview took place (*n* = 4), two because of an illness, two gave no reason.

### 2.2. Data Collection

The interviews lasted approximately 90 min and were held at locations of the participants’ choice or online (e.g., at their home or online by Teams). All interviews were conducted by K.A., M.H., R.G., J.B. and R.H. In all interviews, questions and answers were conveyed verbally, face-to-face. All researchers identified themselves prior to the interviews and gave relevant information on reasons for performing the research, work experience and background.

A semi-structured interview guide was designed by K.A., R.H., J.B. and D.P. (see Appendix A). The use of a semi-structured interview guide ensured that all relevant topics regarding the research questions were discussed. The interview guide started with some short questions about basic demographics such as age and educational attainment: to prevent steering the thoughts of the interviewees, these questions were kept to a minimum. The main topics in the interview guide consisted of McGuire’s five communication factors, and additionally a topic on potential challenges associated with online computer-tailored PA interventions was added. To make this topic more tangible, a specific online computer-tailored PA intervention was used as an example. Details on this intervention are described extensively elsewhere [30,31,32]. All topics were explained in plain language. Images and leaflets were used as examples when questions called for this. Within each topic (i.e., source, message, channel, receiver characteristics and target behavior), about five main questions were formulated. Questions were formulated neutrally, and suggestive questions were avoided. Where possible, the questions from the interview guide were followed with questions such as "can you give an example?" and "is there anything else you can think of?" to ascertain that the information given was as complete and accurate as possible.

Field notes were gathered in which the most relevant findings were noted in order to determine if saturation had been reached. Data collection ended after 39 participants were interviewed: in a meeting between the researchers, the preliminary findings based on field notes were discussed, and as no new information had emerged from the last eight interviews, saturation was deemed to be present, and therefore no new participants were recruited.

All interviews were voice recorded and transcribed ad verbatim, omitting names or other individual identifiers. Transcripts were not returned to participants for comments, and participants therefore did not provide feedback on the findings. The interviews were conducted in Dutch; quotes that are used in this study were translated into English after transcription.

### 2.3. Data Analyses

Data analyses were performed according to the guidelines for thematic analyses [33]. We used a deductive approach in which a preliminary codebook was developed a priori by K.A., R.H., J.B. and D.P. based on McGuire’s framework and our research question. In a deductive approach, codebooks facilitate the exploration of data, the identification of themes and the development of new insights [33]. Key elements of each communication variable were used to define codes. For each code, inclusion and exclusion criteria were formulated and questions from the interview guide linked to the codes were added in the codebook (see Appendix A). The analyses consisted of six iterative steps [33]: in these steps, the codes are generated and reviewed and revised in context of the data if needed [34]. We used Atlas.ti v. 23 as our analytical software [35]. The dataset used for this study is stored in the OSF repository [36].

K.A., M.H. and R.G. coded the interviews. All coding was reviewed by R.H. When R.H. differed in opinion on specific coding, this was discussed with K.A., M.H. and R.G. When no consensus was reached, J.B. or D.P. were included in the discussion. Considering the different backgrounds and experience of the research team, personal reflexivity was sufficiently addressed.

## 3. Results

A total of 39 MAOAs were interviewed, of which 27 were female and 12 were male: gender categorization was based on visual cues or perceived physical characteristics. Age ranged between 50 and 91 years (MD 69.46; SD 7.07). Taking into account the generally accepted age limits for middle-aged (50 through 64) and older adults (65 and older) [37], 14 participants were middle aged and 25 were older adults. For educational attainment, the distribution was 30.77% high, 51.28% medium, 12.83% low and 5.12% unknown. The majority (64.1%) perceived themselves to be sufficiently active in daily life or expressed satisfaction with their amount of PA.

The main results obtained from the interviews describing the perceptions of MAOAs regarding recruitment strategies can be categorized into the five communication factors of McGuire’s theory: in addition to these communication factors, no other themes emerged. For each emerging theme, we added a quote that best reflected the perception of the majority. When perceptions were split, we discussed the differences and added a quote that best reflected the opinion of a large group of the participants [38]

### 3.1. Theme 1: Source

Regarding the source of the recruitment strategies (i.e., the characteristics of the communicator that conveys the message), the consensus from participants was that the general practitioner (GP) and physiotherapist are very suitable sources of information about PA interventions, as they are considered to be trustworthy, reliable and knowledgeable: *“The GP of course, or the physiotherapist, they would know best, if being active is good for you, if you should move more, they can judge that”* (Participant #16).

When inquired about their perception of universities as a source, participants indicated a lack of perceived connection with universities: *“Universities really don’t mean anything to me, I think I just don’t find that appealing, I feel no bond with a university”* (Participant #1).

This observation similarly extended to the local council; despite the legal mandate for Dutch municipalities to promote the well-being of their residents, the large majority of participants appeared unaware of this role and indicated that intervention messages from them would be regarded as patronizing or meddlesome: *“Well, if the municipality uses something to make someone do something, then I’m like: what’s behind this, what’s their underlying intention?”* (Participant #29).

Regarding celebrities as a source of information on recruitment strategies, participants displayed indifference or even expressed concerns that their involvement might have adverse effects on what was intended with the communication. Older adults exhibited less trust in commercial companies like gyms or healthcare insurance companies, whereas this was less pronounced among middle-aged participants: *“Definitely not commercial organizations, because then you would think, well they are making money with this”* (Participant #38).

The participants reported that the more sources with different perspectives, the better, but an overload of information would cause them to feel aversive: *“Yes, repetition is always better of course, because it gives the impression that it is widely supported, from all different corners”* (Participant #25).

### 3.2. Theme 2: Message

In the interviews, we asked participants with what term their age category could best be described in messages about intervention. The term “Fifty-plus”, which is frequently utilized in The Netherlands, was met with mixed reactions; some took the term literally as an indication for their chronological age and therefore found it appropriate, but most participants felt it not fitting, as they perceive a large difference in capabilities between middle-aged and older adults. All participants expressed that there can be large differences between their actual age and how they feel, with one participant saying the following: *“I don’t believe in age, I believe in staying enthusiastic, in joining-in, someone of the age of sixty can be in better shape than someone in the age of thirty”* (Participant #37).

Participants were opposed towards messages that use negative wording and that emphasize physical limitations or chronic diseases. Instead, messages should be framed positively and emphasize that PA interventions help individuals to optimize their well-being and that PA contributes not only to physical well-being but also to mental, cognitive and social health. Words like sports or exercise should be avoided, but the term “PA” is deemed fitting: *“I would prefer that the messages make clear that you can be more independent in your life because of being active, and that your cognition stays intact, and that you can keep doing what you like as long as possible”* (Participant #30).

Regarding motivations to join interventions, it is important according to the participants to highlight the pleasure that PA can trigger: *“A free choice and staging PA as fun, emphasizing that you are just going to enjoy yourself, instead of ‘do you know how important PA is?’… yes of course I know that!”* (Participant #19).

A clear division was seen with participants either leaning to a preference for being active with others or to being active on their own. Those who prefer to be active with others highly value the social interaction that PA can foster, though they do not necessarily need to be deeply involved; even casual conversations are considered sufficient. Those who prefer to be active by themselves emphasized the importance of making clear in the message that the intervention enables this: *“Being active, not just for the reason of being active, but more to have contact with others, and those contacts don’t have to be intensive”* (Participant #29). Another participant stated the following: *“No, that’s not for me, that social aspect, not when it’s sports that I’m looking for”* (Participant #4).

Regarding the appearance of the message, there was a preference for using a striking header and not too much text. When using images like photos they should be of role models that the participants can identify themselves with, i.e., people that are alike in age, gender or physical shape, though some mentioned that diversity and inclusion were also important in images. One middle-aged participant said, upon showing an image of a group of older ladies strolling through a park: “*This makes no sense, people in their fifties are not attracted by this, I only see people older than seventy who are totally out of shape”* (Participant #5).

Images should be realistic, with a type of activity MAOAs generally participate in and pictures from real-life persons (i.e., no off-the-shelf stock photos). Images with people scored better than images with objects, like walking boots.

### 3.3. Theme 3: Channel

Among digital channels, social media was not considered an appropriate channel by older adults, as most of them do not use it. Those that do use it find it not suitable due to its advertisement-heavy nature. In contrast, the middle-aged adults had a more positive attitude toward social media, but expressed doubts whether they would notice a recruitment strategy among all advertisements: *“Well, I’m telling you that Facebook would work counterproductive, because you get so many advertisements that you just don’t look at that sort of messages at all”* (Participant #29).

Another distinct contrast between older and middle-aged adults emerged concerning email. The middle-aged group said they were likely to identify emails as spam, but the older adults found emails useful as a source: *“You have email on your screen, you can read it all immediately”* (Participant #27).

In the printed channels, paper letters were seen as a good channel because people can take the time to read them properly, but MAOAs considered them suitable only when they were addressed personally; otherwise, letters were seen as marketing and were unlikely to be opened. Paper letters or leaflets were found to be appropriate as long as their appearance caught people’s attention sufficiently and they were not combined with other (marketing) leaflets. In The Netherlands, free door-to-door newspapers are a common way for commercial and non-commercial institutions to reach citizens of municipalities; most older adults appreciate the door-to-door newspapers, but the middle-aged adult do not. Some people said that they completely refuse to receive any unaddressed information (by putting a sticker on their letterbox). Paper posters in public spaces received some positive comments, but the location of paper posters seemed to be important: *“It all depends on where you see that poster; if I see it at the local supermarket than I probably would pay less attention to it than if I would see it at a hospital or the physiotherapist.”* (Participant #17).

Word-of-mouth information was in general considered a good channel for information on PA interventions mainly because it comes from someone they know: *“I would be inclined to listen to that and check it out to see if it would also be something for me”* (Participant #24).

Participants did not find mass media appealing because of the likelihood that a campaign would go unnoticed by them. However, national magazines targeting MAOAs were considered useful.

### 3.4. Theme 4: Receiver Characteristics

Participants highlighted the considerable heterogeneity within the MAOA target population. They expressed that people over 50 are very divergent in their needs, wishes and potential for PA, and expressed that a target population for interventions of everyone over the age of 50 is too wide and that a subdivision, by tens for example, is much more appropriate: *“The narrower you can subdivide, the better you will address people, because in all age categories I think you have specific phenomena so to say, when you are older you get worn-out hips and stuff, so it’s important to be as accurate as possible”* (Participant #22).

The middle-aged adults did not find interventions specifically aimed at MAOAs appealing for various reasons, such as already being active enough, wanting to be active with people close to their own age, not feeling "that old" or still having a job and therefore having enough PA. In contrast, some in the middle-aged group express that having a job or an informal care-giver role for parents can hinder them in being sufficiently physically active: *“You have both your family and your work, and the moment your children move out, your parents’ health starts to deteriorate, so you’re always busy with work and with caring”* (Participant #25).

It was striking that a majority of participants expressed themselves to be sufficiently active. However, their answers did not refer to actual guidelines or to what sufficient PA is, but more to them being content with their level of PA. Often, there answers indicated that their perceptions of what actually constitutes sufficient PA were not clear: “*I don’t have to join a PA program; at my age I’m already active enough cleaning my house*” (Participant #5).

The older adults conveyed that younger generations have grown up with a recreational or sports-centric approach to PA and with much more local opportunities for PA, which was less straightforward for their own generation. They also voiced concern about the apparent difficulty in motivating certain individuals to engage in PA: *“If people don’t grow up with PA, you won’t get them to exercise when they are fifty of seventy, you know, at seventy they think ‘I’ve never done it so why start now’”* (Participant #24).

Most participants see PA interventions as more useful for others who they feel are less active than themselves: “*I don’t think I would use an intervention like that at this moment, I don’t need it, but I have quite some people in mind for whom I think it would be really good*” (Participant #11).

A variety of psycho-social personal characteristics are mentioned as reasons to join a PA intervention, such as a lack of self-discipline, the need for a challenge or the social aspect: *“I’m a bit of social sporter, I need to have an appointment with someone, I need that as an incentive to go”* (Participant #19).

A need for autonomy is a reason that is often mentioned not to join a PA intervention: “*I decide how, when, where and which distance I will walk, I can decide for myself what I like, I don’t need an intervention for that*” (Participant #20).

Other incentives that might trigger participants to join a PA intervention were local availability and the ability to seamlessly incorporate it into their daily routines: *“You can exercise of course, but also being active in your daily routines, or taking the bicycle when you do groceries instead of the car”* (Participant #16).

Participants express that the socio-economic status of the receiver is important to take into account, as not everyone can afford to pay for interventions: “*There at the treadmill you pay €45, well then I think, let’s walk around the neighborhood for an hour, that’s for free!*” (Participant #8).

### 3.5. Thema 5: Target Behavior

Middle-aged adults prefer to join the intervention as soon as possible after deciding to do so; this is less important for the older adults. Participants expressed that help options like a telephone line or chat function are desirable. Registration should be as hassle-free as possible: *“Having to make an account and a password, I always find that troublesome, that is a barrier”* (Participant #18).

There is a division among the participants regarding a preference for either online registration or registration by paper forms with pre-paid envelopes, which mainly seemed to depend on the level of computer literacy. Whether the questionnaire should be filled in online or on paper was also largely dependent on the level of computer literacy reported. Middle-aged adults in particular are wary of having to provide a lot of information that is considered personal when registering. MAOAs display a mistrust of computer-tailored advice. Although computer-tailored advice is generally seen as an intriguing technique, most participants express doubts as to whether computer tailoring would really be able to provide advice that is well-suited to the individual, and whether face-to-face contact is not always essential: “*If that advice would say ‘you should go swimming’ then I think, ‘do you know what my hair looks like after swimming?’ so then I would think, ‘thank you for the advice but we’re not going to do that*” (Participant #19). Another participant stated the following: “*It’s good to know that if you become active that there is professional coaching, that you can ask things, like is this good for me?*” (Participant #37).

Computer-tailored interventions often require relatively long questionnaires. In general, participants had no problem with a long questionnaire, as long as they understood the purpose of the large number of questions: “*You always have to invest some time, whether you have to fill in a questionnaire of whether you have an intake in person, that’s inevitable.”* (Participant #18).

An opportunity to try out an intervention is welcomed by MAOAs. Examples of trying out an intervention are seeing the location where the program is held and a trial lesson. In terms of outcome expectancy, participants think being able to obtain insight into what the health benefits of the intervention are may encourage them to take part in the intervention, and also knowing that changes in health will be measured: *“Of course, it is everyone personal’s responsibility but if you give people an insight into how their health has improved, then yes I think that would be the best motivation.”* (Participant #22).

Many participants are interested in interventions that encourage all kinds of PA in their daily life and PA they can perform at home, rather than PA interventions that encourage exercise in an organized setting. However, most participants are concerned that leaving too much responsibility with the individual is not desirable, as they fear that many people do not have the discipline or stamina to perform the activities on their own: *“That can be a drawback, you get some advice, but you still have to do it all by yourself and that is a bottleneck I think”* (Participant #5).

## 4. Discussion

### 4.1. Main Findings

The aim of our study was to identify the preferences of MAOAs regarding recruitment strategies for PA interventions, which we discussed with them according to the communication factors as stated in the model for persuasive communication by McGuire. We will discuss our findings below per communication factors.

### 4.2. Source

The need for the source of recruitment strategies to be trustworthy and knowledgeable was a consistent finding in our study. In particular, GPs and physiotherapists are perceived as such sources, which has been found previously in other research [39,40], especially for those with health issues. This is relevant for our target population, of whom especially the older adults often live with health issues. However, even though the need for prevention in healthcare is receiving more and more governmental and societal support, there are still many practical problems in integrating prevention in primary healthcare [41,42]: for example, in many countries, GPs are heavily overburdened with workload and are therefore not able to integrate advice on interventions into their contacts with their patients. Another barrier is that personal contacts can only reach a small part of the population, as not many people see their GP or physiotherapists regularly. This could implicate that GPs and physiotherapists could better be used as a spokesperson, role model or representative in large-scale communications, for example, an interview with them in a local newspaper or a message from them in a leaflet about an intervention.

Some institutions were found to be less suitable sources as a source of recruitment strategies, such as commercial institutions, universities and municipalities, albeit for different reasons. That municipalities were met with a lack of understanding and even suspicion as a source of recruitment strategies is especially concerning. In The Netherlands, municipalities often play a key role in implementing interventions: by law, the Dutch government has delegated a wide range of public health tasks and responsibilities to municipalities and also the financial means to play an important role in the implementation of health-promotion interventions (including PA). As our participants did not seem to be aware of this role, this implies that if municipalities are used as a source of recruitment strategies, their role should be clearly communicated in the recruitment strategies. The same could apply for other countries where not municipalities but other organizations have an important role in implementing health-promoting (including PA) interventions that the target population is unaware of. Instead of only explaining this in the recruitment strategies, an integral approach can also be recommended. Previous research has shown that in order to have an impact on public health, implementing organizations should work on a more integrated approach, which means that cooperation between different sectors is essential [43]. Such collaboration between implementing organizations could strengthen communication in order to make it clear to the target population why certain organizations contact them with information about health and PA-promoting interventions: a GP, for example, could explain why a municipality is active in the recruitment of PA interventions.

### 4.3. Message

Regarding the message, the need to address such a wide age group with nuanced and varying labels spanning different age brackets was a consistent finding in our study. This does not only apply to labels or wording, but images should also be tailored to smaller segments of the MAOA target population: when images are used in messages, they should be as similar to the target population as possible (in age, vitality and suitable types of PA), which is in line with previous findings [4].

Another important finding is that messages about PA interventions should not focus on health problems or potential diseases; instead, messages should highlight the benefits that PA can have on the broad spectrum of health. This is supported with research on fear appeals [44] and the positive framing of messages: research has shown that this is especially appropriate for older adults [4,45,46].

As there was a clear split in our participants between those who preferred to be active with others and those who preferred to be active alone, messages about interventions should emphasize both options, provided of course that the interventions allow for that. Given the clear split among participants regarding preferences for group versus individual activity, PA interventions for MAOAs should consider incorporating options that allow personal choice in activity settings and types of interventions. The need to adapt interventions to suit the wishes of the target population on major themes has been found previously in other research [11].

### 4.4. Channel

Regarding the channel, there were several issues where a distinction between the middle-aged and older adults seemed more sharply delineated than in the other communication factors. One such issue was email as a channel, which the older adults felt was appropriate, but middle-aged participants expressed that it was likely to be seen as undirected advertising or spam. The matter of confusing recruitment strategies with spam also applied to paper letters as a channel. Therefore, in recruitment strategies there seems to be a demand for personally addressed paper mail, which has been found previously in other research [47]. On the topic of social media, MAOAs did not find it a fitting channel, but for different reasons: the older adults expressed not to use social media, whereas the middle-aged use it but do not find it a useful channel for recruitment strategies, as they would regard a message on social media channels as advertising. This is in line with recent studies that also highlight large differences in social media use between middle-aged and older adults, and that even emphasize that a critical attitude towards the potential of social media to change health behaviors needs to be taken into account [48].

Word of mouth was seen as a good channel, which is consistent with previous research [49]; this may indicate that encouraging people to tell their peers about interventions or giving them incentives for recruiting new participants for an intervention might be something to consider. If this is carried out, it should be noted that peer recruitment can be useful as long as peer recruiters are culturally sensitive and use a personal approach [26].

### 4.5. Receiver Characteristics

Our participants unanimously agreed that the target population for MAOAs is highly heterogenous, and therefore a one-size-fits-all approach is not feasible. In particular, age terminology needs to be used with caution, as most participants did not identify themselves with terms like “people over the age of 50”. This is consistent with research showing that middle-aged people generally feel 10 years younger than their chronological age and older adults feel 14 years younger, and that both middle-aged and older adults have negative attitudes toward people who are older [50,51]. This suggests the need to diversify interventions to better cater to specific subpopulations and apply tailored recruitment strategies and advice for PA interventions for MAOAs, as recommended in previous research [46,52].

Additionally, our study shows that MAOAs present unique challenges in terms of motivating them to be interested in PA interventions. Although prevalence data show that less than half of MAOAs are sufficiently active [53], MAOAs tend to overestimate their level of PA [54,55,56,57]. Similarly, in our sample, the majority of participants perceived themselves to be sufficiently active, but other comments in the interviews suggested that this perception reflected satisfaction with their level of PA rather than actually meeting the guidelines for sufficient PA. Because we did not measure PA objectively, we cannot determine whether this is actually the case. If MAOAs misperceive that they are sufficiently active, it may be more difficult to engage them in PA interventions. Therefore, it seems highly relevant to pay sufficient attention to what constitutes sufficient PA in messages about PA interventions. Older adults may need other forms of motivation, as they expressed that they did not grow up with PA for recreational purposes and therefore seemed to find PA purely for recreation or health purposes less straightforward. For this reason, extra motivational efforts in recruitment strategies are therefore of importance [58,59], such as emphasizing the positive effects that PA can have not only on physical health, but also on cognitive health and general well-being. Albeit for different reasons, both middle-aged and older adults expressed the need for the importance of integrating PA into their daily routines, which is therefore something to address in recruitment strategies.

One receiver characteristics that did not appear to be relevant to MAOA preferences for recruitment strategies was gender: no notable differences were observed between males and females, in contrast to previous research that suggests that males may be harder to reach for PA interventions than females [60]. One explanation for this may be that the majority of our sample was female (69.23%), which could have skewed our findings.

### 4.6. Target Behavior

Our findings showed that the registration process for an intervention should be as straightforward as possible: as privacy regulations often require some administrative effort, it may be advisable to explain why this effort is needed. The registration process should preferably have the option to enlist either online or on paper: the need for these options has been found previously [17,61]. Additional motivating factors included the opportunity to trial the intervention and receiving feedback or monitoring changes in health in the intervention, which is in line with previous research [62].

During the interviews, we presented an online computer-tailored intervention as an example. Computer-tailored interventions are especially known to need elaborate questionnaires. Strikingly, although often lengthy questionnaires are proven to be reasons not to participate or to stop during the registrations process [63,64], our sample was not opposed to lengthy questionnaires, provided that the rationale behind it was clear. A potential explanation may lie in the fact that only 13% of our sample had a low educational attainment: previous research has shown that lower educated individuals find long questionnaires problematic [65,66]. Another issue regarding computer-tailored interventions is that participants struggled to accept that a computer could provide personal and reliable advice, which has been found previously in the literature [67]. This could mean that explaining in recruitment strategies how computer tailoring works is very important.

### 4.7. Strengths and Limitations

A key strength of our study is the larger sample size than what is typical for qualitative research; especially for heterogenous populations, as in our population, a larger sample size is required to provide comprehensive insights [68]. Including five interviewers is also more than average in qualitative studies, which may have limited the possibility of confirmation bias. It should be noted that of our 39 participants, 7 couples were interviewed simultaneously. This may have influenced our findings, as while being interviewed together, the individual participant may have been influenced by the other’s opinion. In order to address this potential issue, we explicitly requested the opinions of both interviewees in the interviews.

While the use of an a priori codebook can potentially introduce a risk of bias, we reviewed and revised the initial codes if needed during the process of thematic analysis: in this way, consistency and a systematic approach was still obtained [33,34,69].

We did not objectively measure the participants’ actual level of PA or ask them if they had any medical conditions that may hinder PA. Whilst we sought to collect data that would give a representative image of this target population, the majority of our participants indicated that they were sufficiently physically active, which contrasts with prevalence data indicating that less than half of this age group is sufficiently active. Although this might suggest that our results are not fully representative, it is known that PA overestimation is quite common [54,55,56,57], and results should be interpreted with this consideration in mind. In addition, it is advisable to check in future research whether participants have a medical condition that may limit physical activity, as this could give more depth to certain perceptions of participants. Gender and age were also not equally represented: 69% of our participants were female, and 64% were older adults. This may have skewed our findings, as previous research has shown gender and age differences in motivational factors for PA [70,71,72,73] and PA intervention participation rates [17,74]. The researchers’ recruitment strategy, which involved recruiting from their own networks and applying snowball sampling, resulted in a greater likelihood of including Caucasian, highly educated participants. Ethnicity was not measured, but the distribution of educational attainment indicates that only 13% had a low educational attainment. As previous research has shown, additional efforts are often required to recruit individuals from low-income populations (of which low education is often a marker) [65,66,75]. Therefore, when generalizing our findings to other populations, it should be taken into account that the distribution of demographic variables in our sample, both known (such as educational attainment) and unknown (such as ethnicity), may mean that the generalization of our findings should be carried out with some caution.

## 5. Conclusions

The current study was successful in identifying factors that can contribute to optimizing recruitment strategies for PA interventions for MAOAs. Most importantly, we recommend including a GP or a physiotherapist as a source of recruitment strategies, possibly in combination with other organizations who play an important role in the implementation of interventions. Furthermore, messages on PA should not focus solely on health problems of potential disease; instead, messages should highlight the benefits that PA can have on the broach spectrum of health and should give information on what levels of PA are deemed sufficient. A major topic that emerged was the heterogeneity of the target population that was visible in all communication factors. This implies a need to tailor recruitment strategies to smaller segments within the overall MAOA population, with age in particular appearing to provide viable cut-off points for such segments. Although not a goal of our study, in addition, some information emerged on what MAOAs find important in PA interventions.

More research on recruitment strategies is essential to better understand how recruitment strategies can be the most effective in attracting MAOAs to PA interventions. Directions for future research could be to obtain more insights in a larger sample, such as exploring the associations between the communication factors as described in our study and demographic characteristics of the target population, in which especially educational attainment or low socio-economic status seems a relevant characteristic. This would allow organizations that design, develop or implement interventions to employ recruitment strategies that are best suited to the needs of the target population, thus optimally using funding to enhance public health.

## Data Availability

The dataset used for this study is stored in the OSF repository [36].

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
