# Peer review of "I Don’t Believe in Age; I Believe in Staying Enthusiastic”: An Exploratory Qualitative Study into Recruitment Strategies Stimulating Middle-Aged and Older Adults to Join Physical Activity Interventions"

_geriatrics, 2024, doi:10.3390/geriatrics9030080_

Round 1

Reviewer 1 Report

Comments and Suggestions for Authors

1-    Page 1, line 20-21: In this sentence, there are two parallel verb actions in this sentence, but the sentence structure is not clear. The sentence should be reorganised to make it clearer, for example: "Despite heterogeneity, MAOA commonly perceive themselves as sufficiently active, indicating a need for improved knowledge on what constitutes sufficient PA, as well as easy enrollment or trying-out interventions."

2-    Page 1, line 29: The words ">50 years" in parentheses and "MAOA" in "(MAOA)" should not appear at the same time, as they imply the same age group. The amendment should read: “Physical activity (PA) in middle-aged and older adults (MAOA) is a topic of major interest and concern for governments worldwide, given its high benefits for public health [1-3].”

3-    Page 2, line 46-48: The sentence "to induce the target population to participate in that intervention" may suggest a potentially manipulative behaviour, which is not in line with general ethical standards because recruitment should be based on voluntary participation. Suggestion for amendment: Replace "induce" with a more appropriate word such as "encourage" or "motivate" to better express voluntary participation. to better express the willingness to participate voluntarily.

4-    Page 2, line 66-67: In this sentence, "change attitudes and actions" seems too absolute, as not all recruitment strategies are designed to change the attitudes and behaviors of the target population. Some recruitment strategies may simply serve to inform or spark interest, rather than directly change attitudes and behaviors. Proposed amendment: change attitudes and actions to something more flexible and objective, for example, "influence attitudes and actions" or "encourage participation".

5-    Page 2, line 81-83: The expression "which image can be strengthened if also a general practitioner is positive about a PA intervention" is not clear enough. It is not clear what "which" refers to, which leads to confusion in sentence structure. Proposed amendment: Change that sentence to something more explicit, as in "the credibility of local municipalities as sources for information on PA interventions can be strengthened if a general practitioner also expresses positivity about a PA intervention".

6-    Page 3, line 143-148: There were 39 participants interviewed, including 7 couples who were interviewed simultaneously. During interview surveys, conducting interviews separately might be more appropriate as the thoughts and answers of the participants could be influenced by each other when interviewed together.

7-    Page 6, line 252: "There was much debate about what term or label should be used to refer to MAOA..." It is mentioned that the definition of "MAOA" varies among individuals, but I think it would be better to add a reference for this statement.

8-    Through Discussion, based on the results section, which is organized by themes, it appears that most participants have individual viewpoints, with few participants sharing the same perspectives. Therefore, I have doubts about whether it is appropriate to draw conclusions in this manner.

9-    Discussion, line 426-428: "That municipalities as a source of recruitment strategies were met with a lack of understanding and even suspicion is especially of concern as in the Netherlands municipalities often play an important role in the implementation of interventions..." The expression in this sentence is not clear enough. Especially "municipalities as a source of recruitment strategies were met with a lack of understanding and even suspicion is especially of concern" part. Proposed amendment: Sentences should be reorganized to make them clearer and more fluid, such as "That municipalities were met with a lack of understanding and even suspicion as a source of recruitment strategies is especially concerning. In the Netherlands, municipalities often play a key role in implementing interventions..."

10- Discussion, line 460-463: "Because this split was so clear, it may even be worth considering whether PA interventions for MAOA should always include some degree of personal choice about whether to be active together or alone, or whether participants should always be given an option for different types of interventions." The expression in this sentence is a little wordy and not concise enough. It can be simplified and restructured. Amendment: It should be expressed more concisely, such as "Given the clear split among participants regarding preferences for group versus individual activity, PA interventions for MAOA should consider incorporating options that allow personal choice in activity settings and types of interventions."

11- Discussion, line 473: "MAOA did not found it a fitting channel..." The expression here is not accurate enough, "did not found" should be changed to "did not find" to make the sentence grammatically correct. Amendment: The sentence should be changed to "MAOA did not find it a fitting channel..."

12- Discussion, line 493-495: The expression here is a little wordy, and unclear, can be simplified and reorganized. Proposed amendment: Can be expressed more concisely. As in "This suggests the need to diversify interventions to better cater to specific subpopulations and apply tailored recruitment strategies and advice for PA interventions for MAOA, as recommended in previous research."

13- Conclusion, line 557-558: "Most importantly, we can recommend including a GP or a physiotherapist as a source of recruitment strategies..." The expression here has a grammatical error, and "recommended" should be changed to "recommend". Proposed amendment: The sentence should be changed to "Most importantly, we recommend including a GP or a physiotherapist as a source of recruitment strategies..."

Comments on the Quality of English Language

There are no further comments regarding the quality of English.

Author Response

Dear reviewer,

We kindly thank you for the time taken to review our manuscript. We feel that by adressing your feedback our manuscript has improved in quality.

Reviewer 1:

We would like to thank the reviewer for the time and attention to our manuscript. We believe that by addressing the points raised, our article has improved in quality.

1- Page 1, line 20-21: In this sentence, there are two parallel verb actions in this sentence, but the sentence structure is not clear. The sentence should be reorganised to make it clearer, for example: "Despite heterogeneity, MAOA commonly perceive themselves as sufficiently active, indicating a need for improved knowledge on what constitutes sufficient PA, as well as easy enrollment or trying-out interventions." Response: We thank the reviewer for this suggestion: we have changed it accordingly.

2- Page 1, line 29: The words ">50 years" in parentheses and "MAOA" in "(MAOA)" should not appear at the same time, as they imply the same age group. The amendment should read: “Physical activity (PA) in middle-aged and older adults (MAOA) is a topic of major interest and concern for governments worldwide, given its high benefits for public health [1-3].” Response: we added “>50 years” to explicitly define what we consider MAOA. We have slightly changed the suggestion by the reviewer, thus addressing the reviewer’s comment, while still including a definition. The new sentence is: “Physical activity (PA) in middle-aged and older adults (MAOA, i.e. adults aged over 50) is a topic of major interest and concern for governments worldwide, given its high benefits for public health [1-3].”

3- Page 2, line 46-48: The sentence "to induce the target population to participate in that intervention" may suggest a potentially manipulative behaviour, which is not in line with general ethical standards because recruitment should be based on voluntary participation. Suggestion for amendment: Replace "induce" with a more appropriate word such as "encourage" or "motivate" to better express voluntary participation. to better express the willingness to participate voluntarily. Response: We have changed ‘induce’ into ‘encourage’.

4- Page 2, line 66-67: In this sentence, "change attitudes and actions" seems too absolute, as not all recruitment strategies are designed to change the attitudes and behaviors of the target population. Some recruitment strategies may simply serve to inform or spark interest, rather than directly change attitudes and behaviors. Proposed amendment: change attitudes and actions to something more flexible and objective, for example, "influence attitudes and actions" or "encourage participation". Response: We have replaced ‘change’ with ‘influence’.

5- Page 2, line 81-83: The expression "which image can be strengthened if also a general practitioner is positive about a PA intervention" is not clear enough. It is not clear what "which" refers to, which leads to confusion in sentence structure. Proposed amendment: Change that sentence to something more explicit, as in "the credibility of local municipalities as sources for information on PA interventions can be strengthened if a general practitioner also expresses positivity about a PA intervention".
Response: Although we agree this is a good amendment, one of the other reviewers suggested to shorten this section (with its examples ). Therefore, we have removed the examples we used there, including this one. We hope the reviewer agrees with the other reviewer that the examples mentioned are not necessary.

6- Page 3, line 143-148: There were 39 participants interviewed, including 7 couples who were interviewed simultaneously. During interview surveys, conducting interviews separately might be more appropriate as the thoughts and answers of the participants could be influenced by each other when interviewed together. Response: We thank the reviewer for this remark and now realize that some words should have been mentioned on this in the Strengths and Limitations. In the interviews we tried to ask as much as possible about how both participants felt about the issues discussed, so we have now added the following text in lines 539: “It should be noted that of our 39 participants, seven couples were interviewed simultaneously. This may have influenced our findings, as while being interviewed together, the individual participant may been influenced by the other’s opinion. In order to address this potential issue, we explicitly requested the opinions of both interviewees in the interviews”.

7- Page 6, line 252: "There was much debate about what term or label should be used to refer to MAOA..." It is mentioned that the definition of "MAOA" varies among individuals, but I think it would be better to add a reference for this statement. Response: We have rewritten these lines to make it clearer that the term mentioned there is not a definition that can be referred to, but rather a term that is commonly used in the Netherlands: we wanted to know what the participants’ perceptions of this term were. The sentences now read as follows: “In the interviews we asked participants with what term their age-category could best be described in messages about intervention. The term “Fifty-plus”, which is frequently utilized in the Netherlands, was met with mixed reactions”.

8- Through Discussion, based on the results section, which is organized by themes, it appears that most participants have individual viewpoints, with few participants sharing the same perspectives. Therefore, I have doubts about whether it is appropriate to draw conclusions in this manner. Response: Our findings are not based on individual viewpoints, but on the viewpoints of the majority of the participants. We have organized our findings per theme, and at each theme we have added quotes of an individual participant that best reflected the viewpoint of the majority. On certain themes, there where clear differences in viewpoints: for example in lines 272-279 or in lines 293-296. We have described these differences to make clear that no common perspective was found. In general, subdivisions in opinions could be made in two groups, such as differences in perceptions of older versus middle aged, or in those wanting to be active with others or alone. In such cases we have added a quote of an individual participant to underline the perception of such a group. We felt that adding more quotes would be overdone, as quotes are merely there to reflect some findings and to bring qualitative content to life. To make this more clear, we have now added the following sentence after the first paragraph of the Results, lines….: “For each emerging theme, we added a quote that best reflected the perception of the majority. When perceptions were split, we discussed the differences, and added a quote that best reflected the opinion of a large group of the participants (Eldh et al., 2020).

9- Discussion, line 426-428: "That municipalities as a source of recruitment strategies were met with a lack of understanding and even suspicion is especially of concern as in the Netherlands municipalities often play an important role in the implementation of interventions..." The expression in this sentence is not clear enough. Especially "municipalities as a source of recruitment strategies were met with a lack of understanding and even suspicion is especially of concern" part. Proposed amendment: Sentences should be reorganized to make them clearer and more fluid, such as "That municipalities were met with a lack of understanding and even suspicion as a source of recruitment strategies is especially concerning. In the Netherlands, municipalities often play a key role in implementing interventions..." Response: We have changed these lines according to the reviewer’s suggestion.

10- Discussion, line 460-463: "Because this split was so clear, it may even be worth considering whether PA interventions for MAOA should always include some degree of personal choice about whether to be active together or alone, or whether participants should always be given an option for different types of interventions." The expression in this sentence is a little wordy and not concise enough. It can be simplified and restructured. Amendment: It should be expressed more concisely, such as "Given the clear split among participants regarding preferences for group versus individual activity, PA interventions for MAOA should consider incorporating options that allow personal choice in activity settings and types of interventions." Response: We have changed these lines according to the reviewer’s suggestion.

11- Discussion, line 473: "MAOA did not found it a fitting channel..." The expression here is not accurate enough, "did not found" should be changed to "did not find" to make the sentence grammatically correct. Amendment: The sentence should be changed to "MAOA did not find it a fitting channel..." Response: We have changed these lines according to the reviewer’s suggestion.

12- Discussion, line 493-495: The expression here is a little wordy, and unclear, can be simplified and reorganized. Proposed amendment: Can be expressed more concisely. As in "This suggests the need to diversify interventions to better cater to specific subpopulations and apply tailored recruitment strategies and advice for PA interventions for MAOA, as recommended in previous research." Response: We have changed these lines according to the reviewer’s suggestion.

13- Conclusion, line 557-558: "Most importantly, we can recommend including a GP or a physiotherapist as a source of recruitment strategies..." The expression here has a grammatical error, and "recommended" should be changed to "recommend". Proposed amendment: The sentence should be changed to "Most importantly, we recommend including a GP or a physiotherapist as a source of recruitment strategies..." Response: We have changed these lines according to the reviewer’s suggestion.

Kind regards,

The authors

Reviewer 2 Report

Comments and Suggestions for Authors

This is an interesting, informative paper.  A few points that will improve its presentation follow:

63. The framework employed certainly helped structure the focus of the paper and the questions that were asked.  It cannot, however, be considered the qualitative approach that was used and this still needs to be explicated.   It appears that a general inductive approach was used and if so, this should be stated later in the paper.

117. Although consent is mentioned, approval of the study by a board charged with protection of human subjects is not mentioned.

125. There is too much detail regarding the research team:  a sentence or two should suffice.

153. Were the online interviews conducted person to person or were they written questions requiring written answers?  

202. A priori codes have some limitations that should be mentioned.  However, it appears that the researchers did not allow themselves to be restricted by this too much.

349. "Physiological" personal characteristics would not include lack of self-discipline. Typo?

390. typo "trial" not "trail"

Author Response

Dear reviewer,

We kindly thank you for the time taken to review our manuscript. We feel that by adressing your feedback our manuscript has improved in quality.

Reviewer 2:

This is an interesting, informative paper. A few points that will improve its presentation follow. Response: We kindly thank the reviewer for the compliments and the time and effort spent to review our article. We feel that by attending to the points mentioned, our article has improved in quality.

63. The framework employed certainly helped structure the focus of the paper and the questions that were asked. It cannot, however, be considered the qualitative approach that was used and this still needs to be explicated. It appears that a general inductive approach was used and if so, this should be stated later in the paper.
Response: We agree with the reviewer that McGuire’s framework provided a solid base for structuring our research. However, the framework in itself is not a qualitative approach, and we did not use it as such. We used McGuire’s framework as a guiding theory in our research. This meant that we based our interview guide and codebook on this theory. In doing so, we used a deductive approach. Thematic analyses using the deductive approach allows for a degree of flexibility by introducing some inductive elements. This means that codes must be generated based on theory, but that they can be reviewed and revised in the context of the data (Braun & Clarke, 2006; DeCuir-Gunby et al., 2011). To address the reviewers’ point, we have now rewritten lines 186-192 as follows: “We used a deductive approach in which a preliminary codebook was developed a priori by KA, RH, JB and DP, based on McGuire’s framework and our research question. In a deductive approach, codebooks facilitate the exploration of data, the identification of themes, and the development of new insights (Braun & Clarke, 2006). Key elements of each communication variable were used to define codes. For each code, inclusion and exclusion criteria were formulated and questions from the interview guide linked to the codes were added in the codebook (see appendix 2). The analyses consisted of 6 iterative steps (Braun & Clarke, 2006): in these steps the codes are generated and reviewed and revised in context of the data if needed (DeCuir-Gunby et al., 2011). These steps are….”

117. Although consent is mentioned, approval of the study by a board charged with protection of human subjects is not mentioned.
Response: The reviewer may have missed that we mentioned this consent, because it was added not in the Methods section (where this consent is often mentioned), but on page 12, line 590-591, where we had to add it to be in line with the journal’s guidelines.

125. There is too much detail regarding the research team: a sentence or two should suffice.
Response: In writing this manuscript, we followed the Consolidated Criteria for Reporting Qualitative Research (Tong et al., 2007). According to these standards, the credentials, occupation, gender, experience and training of the interviewers must be reported. Looking at previous papers published in this journal, it seems that adherence to these criteria are left to the discretion of the author. Therefore, we have now followed the advice of the reviewer and rewritten lines 120-130: “The research team consisted of three principal researchers (RH, JB, DP) supported by three master student researchers (KA, MH, RG). All of the researchers are trained at a masters level in qualitative research. The three main researchers have previous experience in conducting qualitative research.”

153. Were the online interviews conducted person to person or were they written questions requiring written answers?
Response: All interviews were conducted (online) face-to-face; no written questionnaires or responses were used. We have now added the following line to lines 148 – 151: “In all interviews, questions and answers were conveyed verbally, face-to-face.”

202. A priori codes have some limitations that should be mentioned. However, it appears that the researchers did not allow themselves to be restricted by this too much.
Response: There are indeed pros and cons to a priori coding. However, when using deductive thematic analysis in this process, one can begin by formulating initial codes while still leaving some room for review and revision of the coding scheme. In this way, one can adhere to the theoretical framework for answering the research questions (which promotes consistency and a systematic approach) while still reducing potential rigidity and risk of bias by limiting the risk of missing emergent themes (Fereday & Muir-Cochrane, 2006). We agree with the reviewer that some attention to this point is in order. We have now added the following line to the Strengths and Limitations section: “While the use of an a priori codebook can potentially introduce a risk of bias, we have reviewed and revised the initial codes if needed during the process of thematic analysis: in this way consistency and a systematic approach was still obtained (Braun & Clarke, 2006; DeCuir-Gunby et al., 2011; Fereday & Muir-Cochrane, 2006).”

349. "Physiological" personal characteristics would not include lack of self-discipline. Typo?
Response: this is indeed a typo: the word should have been “psycho-social”, which we now have corrected.

390. typo "trial" not "trail"
Response: we have corrected this typo, and thank the reviewer for detecting it.

Kind regards,

The authors

Reviewer 3 Report

Comments and Suggestions for Authors

"I don’t believe in age; I believe in staying enthusiastic": a
qualitative study into recruitment strategies stimulating middle-aged and
older adults to join physical activity interventions

General comments

·       The topic is important and timely.  

·      I don’t think the quote used in the title reflects the paper.

·      The results are crucially significant when presented between groups (male/female, young vs old and especially active /non-active). This approach will effectively reinforce the conclusion that targeted strategies are indeed necessary.

·      Some literature could simply refer readers to the original source. For example, McGuire’s framework takes up too much space in the literature review.

Specific comments.

·      Rephrase Line 35-38 P.1

·      To not avail?

·      Lines 89-91 need scientific support.

·      Gender should be replaced by sex line 139

·      Why recruit in personal and professional networks? This increased the odd of highly educated, white Caucasian individuals.  At least, can you mention the limitations? 

·      I suggest including Lines 165-181 in the appendix.

·      The reference suffice for the last paragraph of the methods

·      64.1% active!!! I thought this paper could identify strategies for those who are inactive.  At the minimum, the results and discussion should be presented separately.

Author Response

Dear reviewer,

We kindly thank you for the time taken to review our manuscript. We feel that by adressing your feedback our manuscript has improved in quality.

Reviewer 3:

General comments ·

The topic is important and timely.
Response: we kindly thank the reviewer for expressing appreciation for our work, and for the time invested in reviewing our paper. ·

I don’t think the quote used in the title reflects the paper.
Response: Looking at the quotes in Theme 2 and Theme 4 of the Results, we believe that the quote used in the title appropriately reflects the participant's view of the reasons for being active or participating in PA interventions, such as the benefits of PA on the broad spectrum of health and being able to do what you enjoy, that PA is fun, that PA enables quality of life etc. A single quote cannot reflect an entire paper, but we believe it reflects an important point of our findings, which we have also elaborated on in the Discussion, i.e., the importance of addressing the right motivations and attitudes toward PA when recruiting this target population for PA interventions. ·

The results are crucially significant when presented between groups (male/female, young vs old and especially active /non-active). This approach will effectively reinforce the conclusion that targeted strategies are indeed necessary.
Response: We agree with the reviewer that presenting results per subgroup can be relevant when there are sufficient differences to warrant such a distinction. However, our analyses showed that only for age, some structurally different perceptions emerged between middle aged vs old: for these age-groups, opinions on several topics were clearly different, and these were all mentioned in our Results. There were no differences found between male/female (which we addressed in line 512-516). The difference between active/non-active participants also did not carry over into other findings, i.e. the perceptions of the active/non-active group was not structurally different on other topics. We therefore feel that presenting results per subgroups is not appropriate. ·

Some literature could simply refer readers to the original source. For example, McGuire’s framework takes up too much space in the literature review.
Response: In light of the reviewer’s advice, we have now shortened lines 73-98. The new text now reads:
“ The communication factors are 1) the source of the communication, i.e the characteristics of the communicator that conveys the message, which comprises the number of sources, unanimity, attractiveness, credibility and demographics; 2) the message factors in the communication, i.e., the type of appeal and information, repetitiveness, inclusion and omission; 3) the channel used to distribute the communication, i.e., modality, directness and context; 4) the receiver of the communication, i.e., demographics, ability, personality and lifestyle, and 5) the target behavior, i.e., immediacy or delay, prevention or cessation, direct/immunization (McGuire, 1989). Combined, the above results in information on who (source) says what (message) how (channel) to whom (receiver) regarding which action (i.e. the target behavior of actually participating in an intervention).”

Specific comments. ·

Rephrase Line 35-38 P.1
Response: we have rephrased this sentence from “When MAOA do not become aware of the existence of an intervention or when they are not triggered to participate in an intervention, the demonstrated effectiveness comes to no avail “ into “If MAOA are not made aware of the existence of an intervention, or if they are not prompted to participate in an intervention, the intervention will not have the impact on PA as previously demonstrated in its efficacy trial.”. ·

To not avail?
Response: We have rephrased this sentence as described above. ·

Lines 89-91 need scientific support.
Response: From lines 79 to 98 we elaborate on the communication factors. We have done this by mentioning elements that can be classified under each communication factor according to McGuire. This information is specified in the source we used. We now see that we have not referred to this source clearly in this part of the text. Therefore we have now added the reference in the rewritten segment on Mc Guire’s framework (lines 84-95). ·

Gender should be replaced by sex line 139
Response: We thank the reviewer for noticing this, and have made this correction. ·

Why recruit in personal and professional networks? This increased the odd of highly educated, white Caucasian individuals. At least, can you mention the limitations?
Response: The research team recruited from their own professional network, anticipating that with 6 researchers involved and the use of snowball sampling, they would have a relatively representative sample of the Dutch population. We concur with the reviewer's assessment that this approach had the potential to attract a highly educated, Caucasian sample. Upon examination of our sample, we observed that the distribution of educational attainment (as mentioned in line 218) was slightly higher than the average for this age group in the Netherlands. We already addressed the latter in the Strengths and Limitations (lines 549-553). In this study, participants' ethnic identification was not explicitly elicited., and therefore we cannot comment on this in the article. In order to address the reviewer’s point, we have now addressed this in the section on strengths and limitations by replacing lines 549-553 with: “The researchers' recruitment strategy, which involved recruiting from their own networks and applying snowball sampling, resulted in a greater likelihood of including Caucasian, highly educated participants. Ethnicity was not measured, but the distribution of educational attainment indicates that only 13% had a low educational attainment. As previous research has shown, additional efforts are often required to recruit individuals from low-income populations (of which low education is often a marker (Chesser et al., 2016; Nicholson et al., 2011; Stuber et al., 2020). Therefore, when generalizing our findings to other populations, it should be taken into account that the distribution of demographic variables in our sample, both known (such as educational attainment) and unknown (such as ethnicity), may mean that generalization of our findings should be done with some caution.”
·

I suggest including Lines 165-181 in the appendix.
Response. As including sample-questions is often used in reporting qualitative research, we decided to include these in our manuscript. When looking at previous qualitative publications in this particular journal, it seems that including sample questions is left to the authors' discretion. We have therefore followed the advice of the reviewer. ·

The reference suffice for the last paragraph of the methods
Response: We have now rewritten this section as follows: “The analysis consisted of 6 iterative steps as described by (Braun & Clarke, 2006). We used Atlas.ti as our analytical software (ATLAS.ti Scientific Software Development GmbH., 2023). The dataset is stored in the OSF repository (Boekhout et al., 2022).” ·

64.1% active!!! I thought this paper could identify strategies for those who are inactive. At the minimum, the results and discussion should be presented separately.
Response: We did not objectively determine whether participants were sufficiently physically active: we asked participants if they thought they were sufficiently active. On this question 64.1% responded that they thought that they were sufficiently active. Most of these responses however did not refer to specific guidelines on what constitutes sufficient physical activity (PA). Instead, they rather reflected a sense of contentment with their own level of PA. During the interviews, it became evident that many participants had a significantly lower idea of what sufficient PA is than what is recommended by the World Health Organization (WHO). Consequently, it is possible that our target population may have the idea they are sufficiently active, when in fact they are not. Upon rereading our manuscript with this reviewer's comment in mind, we concur that we could have made this more clear. As a result, we have made several amendments to our manuscript.
• Lines 218-219 were changed into: The majority (64.1%) perceived themselves to be sufficiently active in daily life or expressed satisfaction with their amount of PA.
• Before the paragraph starting at 339 we added: “It was striking that a majority of participants expressed themselves to be sufficiently active. Their answers however not referred to actual guidelines or to what sufficient PA is, but more to them being contend with their level of PA. Often there answers indicated that their perceptions of what actually constitutes sufficient PA was not clear. “ Quote “I don’t have to join a PA program; at my age I’m already active enough cleaning my house”. (Participant #5).
• Line 500-504 were changed into: “Although prevalence data show that less than half of MAOA are sufficiently active (World Health Organisation, 2022), MAOA tend to overestimate their level of PA (Brenner & DeLamater, 2014; Collombon et al., 2023; Hagstromer et al., 2010; Vandelanotte et al., 2011). Similarly, in our sample, the majority of participants perceived themselves to be sufficiently active, but other comments in the interviews suggested that this perception reflected satisfaction with their level of PA rather than actually meeting the guidelines for sufficient PA. Because we did not measure PA objectively, we cannot determine whether this is actually the case. If MAOA misperceive that they are sufficiently active, it may be more difficult to engage them in PA interventions. Therefore, it seems highly relevant to pay sufficient attention to what constitutes sufficient PA in messages about PA interventions.”
• Lines 541-545 were changed into: “Although this might suggest that our results are not fully representative, it is known that PA overestimation is quite common (Brenner & DeLamater,
2014; Collombon et al., 2023; Hagstromer et al., 2010; Vandelanotte et al., 2011), and results should be interpreted with this consideration in mind.” .

Conclusion: line 562: Messages should also explain what levels of PA are deemed sufficient.
Response: We have now changed these lines to: “Furthermore, messages on PA should not focus solely on health problems of potential disease; instead, messages should highlight the benefits that PA can have on the broach spectrum of health, and should give information on what levels of PA are deemed sufficient.

Kind regards,

The authors

Round 2

Reviewer 1 Report

Comments and Suggestions for Authors

Manuscript ID: geriatrics-2993261-peer-review-v1

Manuscript title: "I don’t believe in age; I believe in staying enthusiastic": A Qualitative Study into Recruitment Strategies Stimulating Middle-Aged and Older Adults to Join Physical Activity Interventions

Major points:

         1-Page 1, Line 50: "Recruitment strategies can be defined..." It seems to refer to some strategies, but what specific strategies are being discussed? It doesn't provide interpretation. In line 22 of the abstract, it also states, "Tailoring recruitment strategies to diverse MAOA segments, based on age, seems crucial for effective engagement."

         2-About the participants: There were 39 participants interviewed, including 7 couples who were interviewed simultaneously. During interview surveys, conducting interviews separately might be more appropriate as the thoughts and answers of the participants could be influenced by each other when interviewed together. It seems that there is no basic participant information regarding the presence of underlying medical conditions. Whether a person has an illness or not can influence their thoughts on physical activity and their motivation for engaging in physical activity.

         3-In the paper, it is written that themes were determined from the answers, but in "3.4 Receiver characteristics," my understanding is that this section should be understood as an evaluation of strategies. Participants are asked about their responses to strategies to promote physical activity. If that's the case, why was it named "Receiver characteristics"

          "There was much debate about what term or label should be used to refer to MAOA..." It is mentioned that the definition of "MAOA" varies among individuals, but I think it would be better to add a reference for this statement.

         4-Page 3,Line 116: Participants. Subheadings need to be repositioned

Author Response

Dear reviewer,

We appreciate your time in reviewing our revised manuscript. Please find attached our response to your valuable remarks. We hope our revision will be deemed satisfactory. 

Kind regards,

The authors

Reviewer 3 Report

Comments and Suggestions for Authors

The revisions were well-responded

Author Response

We thank the reviewer for the time taken to review our manuscript. We are pleased to learn that the reviewer considers our amendments are well performed.